# The CYP80A and CYP80G Are Involved in the Biosynthesis of Benzylisoquinoline Alkaloids in the Sacred Lotus (*Nelumbo nucifera*)

**DOI:** 10.3390/ijms25020702

**Published:** 2024-01-05

**Authors:** Chenyang Hao, Yuetong Yu, Yan Liu, An Liu, Sha Chen

**Affiliations:** State Key Laboratory for Quality Ensurance and Sustainable Use of Dao-Di Herbs, Institute of Chinese Materia Medica, China Academy of Chinese Medical Sciences, No. 16, Nanxiaojie, Dongzhimennei, Beijing 100700, China; chenyanghao98@gmail.com (C.H.); yu040379@gmail.com (Y.Y.); yliu1980@icmm.ac.cn (Y.L.)

**Keywords:** benzylisoquinoline alkaloids, CYP80s, synthetic biology, *Nelumbo nucifera*

## Abstract

Bisbenzylisoquinoline and aporphine alkaloids are the two main pharmacological compounds in the ancient sacred lotus (*Nelumbo nucifera*). The biosynthesis of bisbenzylisoquinoline and aporphine alkaloids has attracted extensive attention because bisbenzylisoquinoline alkaloids have been reported as potential therapeutic agents for COVID-19. Our study showed that *NnCYP80A* can catalyze C-O coupling in both *(R)-N*-methylcoclaurine and *(S)-N*-methylcoclaurine to produce bisbenzylisoquinoline alkaloids with three different linkages. In addition, *Nn*CYP80G catalyzed C-C coupling in aporphine alkaloids with extensive substrate selectivity, specifically using *(R)-N*-methylcoclaurine, *(S)-N*-methylcoclaurine, coclaurine and reticuline as substrates, but the synthesis of C-ring alkaloids without hydroxyl groups in the lotus remains to be elucidated. The key residues of *Nn*CYP80G were also studied using the 3D structure of the protein predicted using Alphafold 2, and six key amino acids (G39, G69, A211, P288, R425 and C427) were identified. The R425A mutation significantly decreased the catalysis of *(R)-N*-methylcoclaurine and coclaurine inactivation, which might play important role in the biosynthesis of alkaloids with new configurations.

## 1. Introduction

Benzylisoquinoline alkaloids (BIAs), are secondary metabolites with diverse structures, which were produced in the order Ranunculales [1] and in the lotus (*Nelumbo nucifera*). Three BIA types, namely monobenzylisoquinoline, aporphine as well as bisbenzylisoquinoline alkaloids, have been detected in the lotus, and some of these compounds have been identified to have a characteristic R-enantiomer conformation [2,3]. In addition, the pharmacological effects of BIAs, such as morphine, extracted from *Papaver somniferum* L. [4] have been widely explored. Neferine, a bisbenzylisoquinoline alkaloid distributed in the lotus plumule, has been reported to be useful for COVID-19 therapy, and therapeutic effects such as anti-inflammatory, antioxidant, antihypertensive, anti-arrhythmic, antiplatelet, antithrombotic, anti-amnesic, and negative inotropic effects have been described [5,6,7]. Nuciferine, a primary aporphine alkaloid found in lotus leaves, is a potentially important candidate for improving hepatic steatosis and managing type 2 diabetes mellitus [8].

*Nelumbo nucifera*, is an ancient aquatic crop used as both medicine and food. However, lotus leaves and seed plumules exhibit different pharmacological effects because of differences in the distribution of aporphine and bisbenzylisoquinoline alkaloids [9,10]. Further elucidation of the molecular mechanisms is critical for the analysis of species evolution and the development of synthetic alkaloids. The critical steps in plant metabolic pathways in microorganisms and the production of natural products are able to be identified owing to the development in metabolic engineering and synthetic biology [11,12,13]. With the development of biotechnology, the biosynthesis of BIAs in the lotus has been revealed to start with norcoclaurine synthase, followed by catalysis by a limited number of enzyme families (*O-*methyltransferase [OMT], NMT, GT, and cytochrome P450s [CYPs]) to generate monobenzylisoquinoline, aporphine and benzylisoquinoline alkaloids. Two OMTs in *N. nucifera* were identified to supplement the synthesis of monobenzylisoquinoline alkaloids [14]. Subsequently, OMTs were isolated and identified, and high-yield mutants were obtained by modification [15]. Using digital gene expression technology, Yang et al. [16] demonstrated that corytuberine synthase participates in the aporphine alkaloid biosynthetic pathway, and *(S)*-*N*-methylcoclaurine and *(S)*-reticuline are the precursors of bisbenzylisoquinoline as well as aporphine alkaloids, respectively. Mechanically wounded lotus leaves display a strong correlation between aporphine alkaloid levels and CYP80G2 in *N. nucifera* [17]. Stereospecificity for (R)-substrates was also demonstrated for two CYP80s in the lotus [18]. Two clustered CYP80 genes were identified to be responsible for the biosynthesis of bisbenzylisoquinoline and aporphine alkaloids based on the common substrate *(S)*-*N*-methylcoclaurine [19]. However, the formation of aporphine and bisbenzylisoquinoline alkaloid skeletons has not been validated and investigated, and the unique R-configured alkaloids and specific aporphine structures in the lotus require further investigation. Therefore, it is critical to investigate the catalytic functions of *Nn*CYP80A and *Nn*CYP80G to elucidate the formation of aporphine and bisbenzylisoquinoline in the lotus.

The monophyletic evolution of BIA biosynthesis in angiosperms indicated that *Nelumbo* was an independent branch but not clustered with *Papaver* and *Ranunculus* [20]. Initially, typical R-enantiomer alkaloids in *N. nucifera* occur primarily in opposition to the S-enantiomers in other species. Second, reticuline, a key precursor of corytuverine, has not been detected in the lotus, suggesting that *N-*methylcoclaurine was used as a substrate to generate aporphine alkaloids in this plant [2,21]. Finally, the formation and coupling of bisbenzylisoquinoline greatly differ between Ranunculales species and the lotus, and the presence of a unique aporphine skeleton lacking the 3′-OH group illustrates the importance of functional studies of *Nn*CYP80A and *Nn*CYP80G [2,22]. CYPs comprise a superfamily of proteins that participate in BIA skeleton formation, and they are associated with hydroxylation, coupling, and isomerization [23]. Berbamunine synthase (CYP80A), which was identified in *Berbaris stolonifera,* can catalyze the formation of bisbenzylisoquinolines (berbamunine and guattegaumerine) from *N-*methylcoclaurine, and the enzyme exhibited configuration selectivity for substrates and reaction rate variance in different expression systems [24]. Corytuberine synthase (CYP80G) was isolated from *Coptis japonica* and heterologously expressed in *Saccharomyces cerevisiae*, and this enzyme produces corytuberine from reticuline via its intramolecular C-C phenol-coupling activity [25]. NMCH (CYP80B), which has hydroxylation activity, is distributed in *P. somniferum, Eschsholzia Californica, C. japonica, Thalictrum flavum,* and *Corydalis yanhusuo* [26].

CYP80A, CYP80G, and CYP80B have also been annotated in *N. nucifera* based on sequence similarity with BIA biosynthetic genes in Ranunculales species, and they have been hypothesized to catalyze the synthesis of bisbenzylisoquinoline and aporphine alkaloids in the lotus. In this study, we expressed *Nn*CYP80A and *Nn*CYP80G in *S. cerevisiae*, verified their enzymatic function, explored the molecular catalytic mechanism of *Nn*CYP80G, and further improved the synthesis of alkaloids in the lotus.

## 2. Results

### 2.1. Cloning and Identification of CYP80 from N. nucifera

The CYP80A and CYP80G genes, which are annotated as *(S)-N*-methylcoclaurine 3′-hydroxylase in the transcriptome data of *N. nucifera* were cloned from the plumules and leaves of *N. nucifera* and named *Nn*CYP80A and *Nn*CYP80G, respectively. *Nn*CYP80A contains 1461 nucleotides encoding 487 amino acids, whereas *Nn*CYP80G contains 1458 nucleotides, encoding 486 amino acids. An assessment of the conserved domains of *Nn*CYP80A and *Nn*CYP80G revealed the presence of conserved CYP regions at amino acids 57–482 and 57–481, respectively.

The biosynthesis of BIAs is mainly controlled by three CYP subfamilies, namely CYP719, CYP82, and CYP80, each of which is aggregated in a cluster (Figure 1A). CYP80s were clustered together in a clade that included *Bs*CYP80A, which catalyzes the C-O coupling reaction [24]. *Cj*CYP80G was identified as a C-C coupling enzyme that can catalyze the formation of *(S)-*corytuberine from *(S)-N*-methylcoclaurine, and three potential corytuberine synthases were also identified in *Podophyllum peltatum, Sinopodophyllum hexandrum,* and *Thalictrum thalictroides*. CYP80B has been characterized to catalyze the production of *(S)-*3′-hydroxy-*N-*methylcoclaurine from *(S)-N*-methylcoclaurine, and this CYP is also annotated as NMCH. *Nn*CYP80A and *Nn*CYP80G displayed 88.89% amino acid sequence identity. In addition, copies of these CYPs have been reported in other species [24,25]. *NnCYP80B2* exhibited 57.36% sequence identity with *CyNMCH*.

Transcriptome databases of different parts of *N. nucifera*, including the leaf stalk (LS), flower stalk (FS), new root (NR) and others, were constructed using Illumina paired-end sequencing technology. CYP80A and CYP80G expression significantly varied in different tissues of *N. nucifera* based on the transcriptome databases (Figure 1B). The results showed that *Nn*CYP80G expression was high in all examined tissues excluding PT and LS, as determined by fragments per kilobase of transcript per million mapped reads. *NnCYP80A* expression was higher in reproductive tissues (P, PT, O, PS and SM) and lower in the FSL, LSH and LBP. The results highlighted the specificity of CYP80 A and CYP80G expression.

### 2.2. Functional Characterization of NnCYP80A

After recombinant *Nn*CYP80A was transferred into yeast cells, its expression was induced with galactose, and microsomes were extracted for in vitro assays using **4** and **5** as the substrates. UPLC-Q-TOF-MS/MS demonstrated that *Nn*CYP80A could catalyze the production of three different bisbenzylisoquinoline alkaloids ([M + H]^+^ *m/z* 597.29) from *N*-methylcoclaurine (Figure 2), validating that this gene could generate three C–O linkages. Peak 2 was the main product only when compound **4** was the substrate. The major peaks observed in the ESI-MS spectra were [M + H-C_19_H_22_NO_3_]^+^ at *m/z* 312.1582 and [M + H-C_12_H_15_NO_2_]^+^ at *m/z* 205.1089, which were suggestive of 8-C and 13′-O coupled (Figure 2A). Using compound **5** as the substrate, the most abundant peak was peak 3, which was presumptively identified as nelumboferine. Regarding the catalysis reaction, a C–O bond might have formed at 7-O and 12′-C (Figure 2B). When adding the mixture substrate (4:5, 1:1) as substrates, peak 1, the main catalysate bridge connection at 13-O and 14′-C, which was identified as guattegaumerine, has been improved by the authentic standard (Figure 2C).

### 2.3. Functional Characterization of NnCYP80G

*Nn*CYP80G was cloned and tested via recombinant expression in *S. cerevisiae*. *N*-methylcoclaurine is regarded as the key precursor of aporphine alkaloids in the lotus. Compounds **4** and **5** were added in a reaction system with NADPH. UPLC-Q-TOF-MS/MS revealed a [M-2H]^+^ ions at *m/z* 298.1428, 284.1281, and 328.1543, indicating that *Nn*CYP80G could induce C-C phenol-coupling in compound **4**, **5**, **6**, and **8** (Figure 3A). The product generated from compound **4** and **5** was determined to have the molecular formula C_18_H_19_NO_3_, and based on a series of fragment ions [M + H-CH_3_]^+^ at *m/z* 283.1146, the product was identified as sparsiflorine with two configurations based on the control and mass spectrum fragmentation (Figure 3B). Sparsiflorine, which has no commercially available standard, was previously reported in *Croton sparsiflorus* [27]. Meanwhile, compound **6**, which is unmethylated at the N-position, was used for functional characterization. Compared to the control, the product had the molecular formula C_17_H_17_NO_3_, and based on fragment ions [M + H]^+^ at *m/z* 284.1281, the product was identified as coclaurine (Figure 3B). *Nn*CYP80G could catalyze **8** to form corytuberine, which had fragment ions [M + H]^+^ at *m/z* 328.1469. The reaction efficiency of the *Nn*CYP80G microsomal fraction increased in the order of **4** > **8** > **5** > **6**. No product was detected using compound **7** as the substrate.

### 2.4. Mutational Analysis of C-C Coupling by NnCYP80G

To clarify the enzymatic mechanism of the C-C coupling activity of *Nn*CYP80G, site-directed mutagenesis was performed using protein modeling with substrate docking and amino acid sequence alignment between *Cj*CYP80G and *Nn*CYP80G. The model of *Nn*CYP80G was generated using Alaphfold 2. Six sites (GLY39, GLY69, ALA211, PRO288, ARG425, and CYS427) were chosen for mutagenesis (Figure 4A). Then, compounds **4**, **5**, **6** were docked into the model, followed by sequence alignment (Appendix A). The catalytic activity of the mutants was screened using microsomes. Compared to the activity of wild-type *Nn*CYP80G, the mutants G39A, G69A, A211E, P288A, R425A, and C427A exhibited reduced catalytic activity for all three substrates (Figure 4B). Interestingly, R425 was identified as a key residue, as mutation of this residue resulted in remarkably decreased catalytic activity for compound **5** and the complete loss of activity for compound **6** (Figure 4B). The isomer selectivity of the mutant is of great significance for the biosynthesis of alkaloids.

## 3. Discussion

Previous research revealed that CYP80B hydroxylates *(S)-N*-methylcoclaurine to produce *(S)*-3′-hydroxy-*N*-methylcoclaurine in *E. californica* and *C. yanhusuo* [26,28]. No catalytic ability was detected for *Nn*CYP80B using either *(S)-N*-methylcoclaurine or *(R)-N-*methylcoclaurine as the substrate. UPLC-QTOF-MS/MS only detected *(S)*-3′-hydroxy-*N*-methylcoclaurine in the reference standard; however, no product was formed (Figure 5A). *Nn*CYP80B was cloned from *N. nucifera* which lacks the ability to synthesize 3′-hydroxy-*N*-methylcoclaurine and reticuline. Given that no such compounds have been reported in the lotus, we predict that the function of *Nn*CYP80B is indeed lost in this plant.

The reaction mechanism of CYP80G is considered a biradical process, in which the 3′-H atom of the C-ring and the 7-hydroxy group of the A-ring are catalyzed to generate two radicals and achieve biradical coupling [29]. A previous report [25] using *C. japonica* revealed that the catalytic efficiency of *Cj*CYP80G is lowest for *(S)-N*-methylcoclaurine and *(S)*-coclaurine. Conversely, *Nn*CYP80G has significantly higher catalytic efficiency for *(S)-N*-methylcoclaurine and *(R)-N*-methylcoclaurine, and its catalytic efficiency for coclaurine is also higher. To provide molecular support for the detection of aporphine alkaloids in the lotus, it is suggested that the origin of genes should be considered when constructing engineered strains in synthetic biology.

*N. nucifera* is an ancient basal angiosperm (Figure 5B) with an extremely specific enzymatic function. Recently, a phylogenomic study demonstrated that BIA biosynthesis is of monophyletic origin in Ranunculales, but it arose independently in Proteales [30]. However, the BIA biosynthetic enzymes of the sacred lotus have not been characterized. Meanwhile, the efficiency of the aforementioned catalytic reactions is extremely low, highlighting the need for further optimization to increase yield. Furthermore, some unidentified products need to be further identified by nuclear magnetic resonance spectroscopy. In this study, we improved the synthesis pathways of bisbenzylisoquinoline and aporphine alkaloids in *N. nucifera*, verified the functions of CYP80A and CYP80G, and probed the similarities and differences between BIA biosynthesis mediated by CYP80A and CYP80G in Ranunculales. A more explicit framework for the subsequent study of the biosynthesis of BIAs is presented in Appendix A, making it possible to produce medical BIA compounds in N. nucifera via heterologous synthesis.

## 4. Materials and Methods

### 4.1. Plant Materials and Chemicals

The leaves and plumules of *N.nucifera* cultivar ‘HuangMuDan’ were obtained and subsequently frozen in liquid nitrogen immediately. Regarding the standards, *(S)-N*-methylcoclaurine (**4**) was purchased from Wuhan Qiongge Biotechnology Co., Ltd. (Wuhan, China), and *(R)-N*-methylcoclaurine (**5**) was purchased from BioBioPha Co., Ltd. (Kunming, China). Both standards were verified via circular dichroic chromatography (Appendix A). Coclaurine (**6**) was purchased from Sichuan Victory Biological Technology Co., Ltd. (Chengdu, China). Armepavine (**7**) was purchased from Jiangxi Baicaoyuan Bio-technology Co., Ltd. (Nanchang, China). Reticuline (**8**) was purchased from Shanghai Yuanye Bio-technology Co., Ltd. (Shanghai, China). Guattegaumerine (**1**) was obtained from Chengdu DeSiTe Biological Technology Co., Ltd. (Chengdu, China). *(S)*-3′-hydroxy-*N*-methylcoclaurine (**9**) was obtained from Toronto Research Chemicals Inc. (North York, ON, Canada). NADPH was commercially purchased from Shanghai Macklin Biochemical Co., Ltd. (Shanghai, China). Other analytical solvents, including formic acid, acetonitrile, and methanol used in UPLC-MS/MS and additive solvents, were obtained from Sigma-Aldrich (St. Louis, MO, USA).

### 4.2. Gene Cloning and Enzyme Expression in Yeast

Total RNA of 12 lotus tissues from LS, FS, NR, LBP, RF, LSH, LP, PT, FSL O, PS, and SM were extracted using an Easy Fast Plant Tissue Kit (Tiangen Biotech, Beijing, China). RNA integrity was assessed using an RNA Nano 6000 Assay kit on an Aglient bioanalyzer 2100 system (Agilent Technologies, Palo Alto, CA, USA). Therefore, illumine paired-end sequencing technology was used.

Total RNA was reverse-transcribed to cDNA using a FastKing RT Kit with gDNase (Tiangen Biotech). Using a T4 DNA Ligase (TransGen Biotech, Beijing, China), the genes (generated by PCR using the primers listed in Appendix A) were connected to the eukaryotic expression vector pESC-CPR-His, and were transformed into *Escherichia coli* Trans T1 (TransGen Biotech), which was sent to SinoGenoMax Co., Ltd. (Beijing, China) for sequencing. The recombinant plasmid was transformed into yeast strain WAT11, and the control strain was transformed with the empty vector to achieve heterologous expression. Both strains were cultured in SD-His medium at 30 °C for 48 h. A single colony was grown for 48 h. The cells were centrifuged at 1000× *g* for 5 min and resuspended in YPDE (10 g L^−1^ yeast extract, 10 g L^−1^ tryptone, 20 g L^−1^ glucose, 3% ethanol) for cultivation for 48 h when glucose was exhausted. Approximately 3% galactose and 2.5% ethanol were added for induction. Afterwards, the microsomes were extracted. Cells were dispersed at 4 °C using a low-temperature and ultrahigh-pressure nanomaterial preparation and dispersion machine. After centrifugation at 9000× *g* for 40 min, microsomes were precipitated by differential centrifugation at 120,000× *g*. Pellets were resuspended in TEGM (500 μL of Tris-HCl [1 M, pH 7.5], 20 μL of EDTA [0.5 M, pH 8.0], 4 mL of 50% glycerinum, 1 μL of β-mercaptoethanol, and sterile water added to 10 mL.

CYP80A activity was assessed in a reaction system containing 200 μL of microsomal protein, 8.4 μL of NADPH (100 mg mL^−1^), 3.4 μL of substrate (10 μM), 10 μL of NaCl (2.5 M), and 50 μL of tricine buffer (pH 7.5). CYP80G and CYP80B activities were assessed in tricine buffer (pH 7.6). The reaction mixture was incubated at 30 °C and 200 rpm for 16 h, and HCl was added to stop the reaction and precipitate proteins. Nitrogen was used for enrichment, followed by extraction with 200 μL of methyl alcohol.

### 4.3. LC–MS Analysis

Catalyzed production analysis was performed using UPLC-Q-TOF-MSn (1290 photodiode array and a 6450 triple quad mass time-of-flight mass spectrometer instrument, Agilent Technologies, Palo Alto, CA, USA) and a chromatographic column (Waters, Milford, MA, USA, ACQUITY UPLC BEH C18, 2.1 mm × 100 mm, 1.7 μm). The mobile phase consisted of 0.1% formic acid (A) and acetonitrile (B), and the sample injection volume was 2 μL. The linear gradient elution program was as follows: 5–95% B from 0 to 15 min; flow rate, 0.3 mL min^−1^; and temperature, 35 °C.

The ESI source was set to the positive ionization mode, and for the Q-TOF-MS detector: nebulizer, 45 psig, nozzle voltage, 500 V; Vcap, 4000 V; sheath gas temperature, 350 °C; drying gas flow, 8 L min^−1^; sheath gas flow, 11 L min^−1^; collision energy, 30 eV; and scan range, *m/z* 100–1500 Da. The measured masses were modified using internal references (purine and HP-0921) in real time.

### 4.4. Modeling Docking and Mutagenesis

A structural model for elucidating the catalytic mechanism of CYP80G was constructed using Alphafold 2. The substrate was docked into the structure using Autodock Tools (1.5.7). Molecular distances were calculated using PyMol (2.5.3). Substitutions for the selected residues in CYP80G were generated by PCR using the primers listed in Appendix A.

## 5. Conclusions

We verified the function of three CYP80 genes in lotus and demonstrated functional differentiation in monobenzylisoquinoline, aporphine, and bisbenzylisoquinoline alkaloids. Among them, both *Nn*CYP80A and *Nn*CYP80G show strong conformational selectivity, and *Nn*CYP80A shows functional differences in several species, while *Nn*CYP80G may mediate a new alkaloid synthesis, which is specifically distributed in lotus species. This study provides a new perspective for the biosynthesis of alkaloids, which is helpful to further improve the synthesis of aporphine and bisbenzylisoquinoline alkaloids.

## Figures and Tables

**Figure 1 ijms-25-00702-f001:**
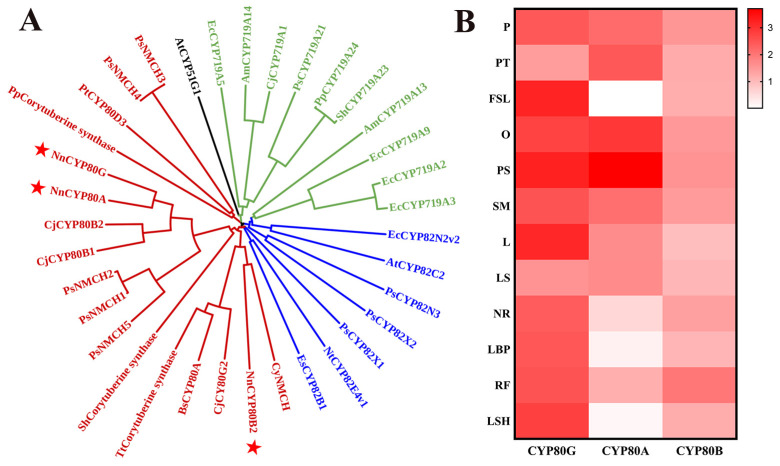
NJ phylogenetic tree of functional characterized P450s involved in the biosynthesis of BIAs (**A**) and expression heat map of *Nn*CYP80s in 12 organs of *N. nucifera* (**B**). CYP80s isolated from *N. nucifera* is represented by red star. The 12 organs are: pistil (P), petal (PT), flower sepal (FSL), ovary (O), petalized stamen (PS), stamen (SM), leaf (L), leaf stalk (LS), new root (NR), leaf bud primordium (LBP), root fibril (RF), leaf sheath (LSH).

**Figure 2 ijms-25-00702-f002:**
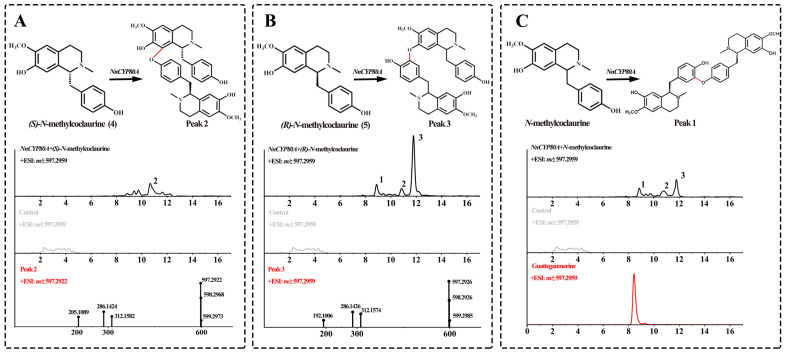
Substrates **4** and **5** catalyzed by NnCYP80A. Reaction and extracted ion chromatograms (EICs) of *m/z* 300.1594 and 597.2959 with *(S)-N*-methylcoclaurine (**A**), *(R)-N*-methylcoclaurine (**B**) and *N-*methylcoclaurine (**C**) as the substrate. The red line indicates reaction generation.

**Figure 3 ijms-25-00702-f003:**
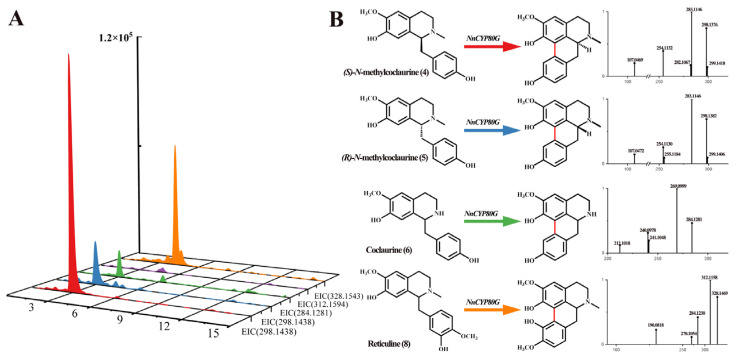
Substrates **4**, **5**, **6**, **7** and **8** catalyzed by *Nn*CYP80G (**A**) and mass spectrum of product (**B**). Reaction and extracted ion chromatograms (EICs) of *m/z* 298.1438 with compounds **4**, **5** as the substrate. Reaction and extracted ion chromatograms (EICs) of *m/z* 284.1281 with compounds **6** as the substrate. Reaction and extracted ion chromatograms (EICs) of *m/z* 312.1594 with compounds **7** as the substrate. Reaction and extracted ion chromatograms (EICs) of *m/z* 328.1543 with compounds **8** as the substrate. The read line indicates reaction generation.

**Figure 4 ijms-25-00702-f004:**
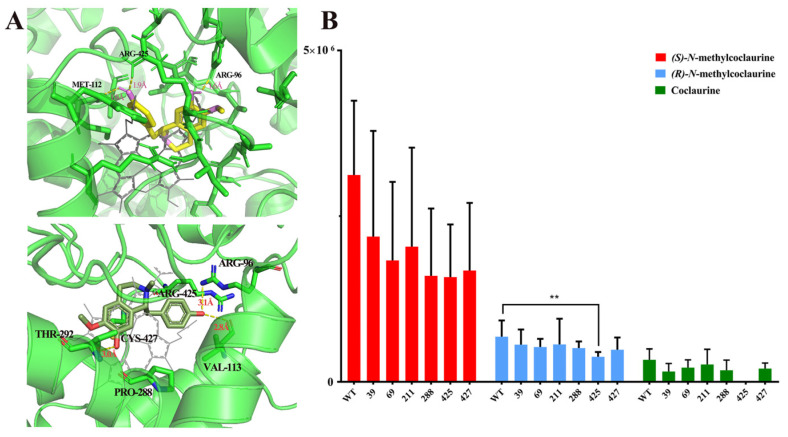
Docking result for compounds **4** and **5** in the *Nn*CYP80G model, with side chains for the targeted residues shown (**A**). Functional screening of mutants with compounds **4**, **5**, **6** as substrates (**B**). The yellow structure is *(S)-N*-methylcoclaurine, the dark green structure is *(R)-N*-methylcoclaurine, the bright green is the protein structure, and the gray structure is the ligand in (**A**). ** *p* < 0.01.

**Figure 5 ijms-25-00702-f005:**
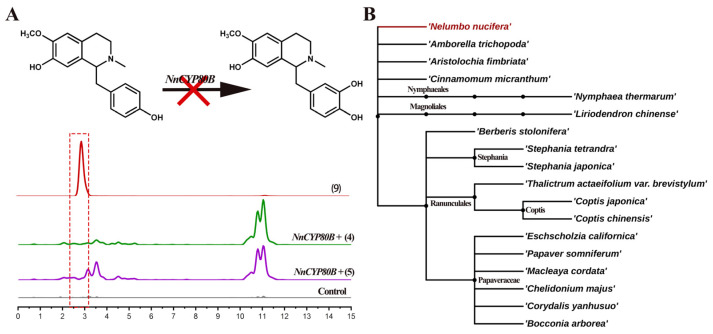
Substrates **4**, **5** catalyzed by *Nn*CYP80B. Reaction and extracted ion chromatograms (EICs) of *m/z* 300.1594 and 316.1543 with *N*-methylcoclaurine as the substrate (**A**). Phylogenetic relationship of eighteen analyzed plant species, adapted from Angiosperm Phylogeny Group IV (APG IV) from the botanical classification system (**B**).

## Data Availability

Data is contained within the article.

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
