# Peer review of "The CYP80A and CYP80G Are Involved in the Biosynthesis of Benzylisoquinoline Alkaloids in the Sacred Lotus (Nelumbo nucifera)"

_ijms, 2024, doi:10.3390/ijms25020702_

Round 1

Reviewer 1 Report

Comments and Suggestions for Authors

The research topic is very interesting and actual, the presented results are original. However, the manuscript needs to be revised before publishing.

The title should be corrected - "the enzymes are catalyzed" is not a correct expression.

The section Material and methods needs an extensive revision. The transcriptomic data from different tissues are mentioned but there is no information about the construction of sequencing library, downstream in silico analysis, differential analysis, etc. or accession number of available data in NCBI. It is not clear whether authors reffer to the data acquired for this study or already published data.

How was the site mutagenesis performed? Why did the authors choose the mentioned six aminoacids? Please mention the substrates used for molecular docking in Material and methods and cite the docking software. 

The functional screening of mutants showed ambiguous results. It can not be concluded that mutants exhibited significantly decreased activity. The error bars are quite high. 

Fig. A1 Please correct the number of figure and explain the presented alignment. The mutagenesis sites could be highlited as well.

Fig. A2 Please improve the resolution if possible. Zooming will result in the obvious artifacts around the letters.

Supplementary data: Table 1 contains solely DNA sequences, please correct the caption. The sequences are available in NCBI database and have the accession numbers. Please consider omitting them from Supplementary material.

Author Response

Thank you for your letter about Manuscript and for the reviewers’ helpful comments concerning our manuscript entitled “The CYP80A and CYP80G are involved in the biosynthesis of benzylisoquinoline alkaloids in the sacred lotus (Nelumbo nucifera)”. We have studied your comments carefully and have made corrections which hope meet with their approval. 

 Point-by-point response to Comments and Suggestions for Authors

Comments 1: The title should be corrected - "the enzymes are catalyzed" is not a correct expression.

Response 1: Thank you for pointing this out. I have revised it.

Comments 2: The section Material and methods needs an extensive revision. The transcriptomic data from different tissues are mentioned but there is no information about the construction of sequencing library, downstream in silico analysis, differential analysis, etc. or accession number of available data in NCBI. It is not clear whether authors reffer to the data acquired for this study or already published data.

Response 2: Thank you for correcting this. I have added the details in the section Material and methods.

Comments 3: How was the site mutagenesis performed? Why did the authors choose the mentioned six aminoacids? Please mention the substrates used for molecular docking in Material and methods and cite the docking software.

Response 3: Thank you for your question. We selected these six sites based on the results of homologous sequence alignment and modeling docking, but because the above are based on speculation, only six sites were tried for mutation.

Comments 4: The functional screening of mutants showed ambiguous results. It can not be concluded that mutants exhibited significantly decreased activity. The error bars are quite high.

Response 4: Thank you for your suggestion. The statistical analysis of mass spectrum peak areas will lead to high error bar, but it is difficult to correct its content by standard curve because of the low efficiency of enzyme activity. In the interpretation of the results, we only confirmed the inactive sites, and the rest sites were speculated, pending further verification.

Comments 5: Fig. A1 Please correct the number of figure and explain the presented alignment. The mutagenesis sites could be highlighted as well.

Response 5: Thank you for correcting this. I have revised the number of Fig. S1. The mutagenesis sites are highlighted by the yellow box.

Comments 6: Fig. A2 Please improve the resolution if possible. Zooming will result in the obvious artifacts around the letters.

Response 6: Thank you for your suggestion. I have improved the Fig. S2 resolution by 600 dpi.

Comments 7: Supplementary data: Table 1 contains solely DNA sequences, please correct the caption. The sequences are available in NCBI database and have the accession numbers. Please consider omitting them from Supplementary material.

Response 7: Thank you for your correction, we have omitted it.

Reviewer 2 Report

Comments and Suggestions for Authors

The manuscript is interesting and valuable, generally well written and well prepared. The applied methods are modern, the results are clearly presented.

The knowledge on the biosynthesis of specialized metabolites, including alkaloids, are of great importance in biotechnology.

I have a serious doubt about the title of this manuscript “The CYP80A and CYP80G are catalyzed in the biosynthesis of benzylisoquinoline alkaloids in the sacred lotus (Nelumbo nucifera)”. In my opinion this title should be corrected. CYP80A and CYP80G are the enzymes. So they are not “catalyzed”, but they are catalysts (biocatalysts).

The possible title could be:

“The CYP80A and CYP80G are involved in the biosynthesis of benzylisoquinoline alkaloids in the sacred lotus (Nelumbo nucifera)”.

If to use “catyzed” – that it should be written that ”The biosynthesis of benzylisoquinoline alkaloids in the sacred lotus (Nelumbo nucifera) is catalyzed by the CYP80A and CYP80G”

But this sentence is not true, because there are also other enzymes involved in this biosynthetic pathway.

Other small remarks:

Lines 57, 127. „N. nucifera” should be in italics

Line 69. “The monophyletic evolution of BIA biosynthesis in angiosperms was presumed to be that Nelumbo was an independent branch compared with Papaver and Ranunculus”. This sentence is awkward and difficult to understand, what means “presumed to be that Nelumbo was”?

Author Response

Thank you for your letter about Manuscript and for the reviewers’ helpful comments concerning our manuscript entitled “The CYP80A and CYP80G are involved in the biosynthesis of benzylisoquinoline alkaloids in the sacred lotus (Nelumbo nucifera)”. We have studied your comments carefully and have made corrections which hope meet with their approval. 

 Point-by-point response to Comments and Suggestions for Authors

Comments 1: I have a serious doubt about the title of this manuscript “The CYP80A and CYP80G are catalyzed in the biosynthesis of benzylisoquinoline alkaloids in the sacred lotus (Nelumbo nucifera)”. In my opinion this title should be corrected. CYP80A and CYP80G are the enzymes. So they are not “catalyzed”, but they are catalysts (biocatalysts).

Response 1: Thank you for pointing this out. I have revised the title of manuscript “The CYP80A and CYP80G are involved in the biosynthesis of benzylisoquinoline alkaloids in the sacred lotus (Nelumbo nucifera)”

Comments 2: Lines 57, 127. „N. nucifera” should be in italics.

Response 2: Agree. I have revised it.

Comments 3: Line 69. “The monophyletic evolution of BIA biosynthesis in angiosperms was presumed to be that Nelumbo was an independent branch compared with Papaver and Ranunculus”. This sentence is awkward and difficult to understand, what means “presumed to be that Nelumbo was”?

Response 3: Thank you for your suggestion. I have revised it.

Round 2

Reviewer 1 Report

Comments and Suggestions for Authors

The manuscript was precisely revised and deserves to be published.